# Problematic use of the internet, smartphones, and social media among medical students and relationship with depression: An exploratory study

Jonathan Sserunkuuma[1], Mark Mohan Kaggwa[1,2,3], Moses Muwanguzi[2]*, Sarah Maria Najjuka[4], Nathan Murungi[1], Jonathan Kajjimu[1], Jonathan Mulungi[1], Raymond Bernard Kihumuro[1], Mohammed A. Mamun[5,6], Mark D. Griffiths[7], Scholastic Ashaba[1,2]

1 Faculty of Medicine, Mbarara University of Science and Technology, Mbarara, Uganda, 2 Department of Psychiatry, Faculty of Medicine, Mbarara University of Science and Technology, Mbarara, Uganda, 3 Department of Psychiatry and Behavioural Neurosciences, McMaster University Hamilton, Hamilton, Ontario, Canada, 4 College of Health Sciences, Makerere University, Kampala, Uganda, 5 CHINTA Research Bangladesh, Savar, Dhaka, Bangladesh, 6 Department of Public Health and Informatics, Jahangirnagar University, Savar, Dhaka, Bangladesh, 7 Psychology Department, Nottingham Trent University, Nottingham, United Kingdom

* muwanguzimoses04@gmail.com

## Abstract

### Background

Students in sub-Saharan African countries experienced online classes for the first time during the COVID-19 pandemic. For some individuals, greater online engagement can lead to online dependency, which can be associated with depression. The present study explored the association between problematic use of the internet, social media, and smartphones with depression symptoms among Ugandan medical students.

### Methods

A pilot study was conducted among 269 medical students at a Ugandan public university. Using a survey, data were collected regarding socio-demographic factors, lifestyle, online use behaviors, smartphone addiction, social media addiction, and internet addiction. Hierarchical linear regression models were performed to explore the associations of different forms of online addiction with depression symptom severity.

### Results

The findings indicated that 16.73% of the medical students had moderate to severe depression symptoms. The prevalence of being at risk of (i) smartphone addiction was 45.72%, (ii) social media addiction was 74.34%, and (iii) internet addiction use was 8.55%. Online use behaviors (e.g., average hours spent online, types of social media platforms used, the purpose for internet use) and online-related addictions (to smartphones, social media, and the internet) predicted approximately 8% and 10% of the severity of depression symptoms,

**Data Availability Statement:** All data are available from: https://doi.org/10.6084/m9.figshare. 22794644.

**Funding:** The author(s) received no specific funding for this work.

**Competing interests:** The authors have declared that no competing interests exist.

respectively. However, over the past two weeks, life stressors had the highest predictability for depression (35.9%). The final model predicted a total of 51.9% variance for depression symptoms. In the final model, romantic relationship problems ($\beta = 2.30$, $S.E = 0.58$; $p<0.01$) and academic performance problems ($\beta = 1.76$, $S.E = 0.60$; $p<0.01$) over the past two weeks; and increased internet addiction severity ($\beta = 0.05$, $S.E = 0.02$; $p<0.01$) was associated with significantly increased depression symptom severity, whereas *Twitter* use was associated with reduced depression symptom severity ($\beta = 1.88$, $S.E = 0.57$; $p<0.05$).

## Conclusion

Despite life stressors being the largest predictor of depression symptom score severity, problematic online use also contributed significantly. Therefore, it is recommended that medical students' mental health care services consider digital wellbeing and its relationship with problematic online use as part of a more holistic depression prevention and resilience program.

## Introduction

According to the International Telecommunication Union (2022), there was a 17% increase in internet use globally during the first two years of the COVID-19 pandemic [1]. A number of factors have been posited to explain the growth, such as frequent engagement in telework, teleconferencing, online learning (e-learning), telehealth, and online shopping that were engaged in as a way of adhering to spatial distancing, a restrictive measure to inhibit the spread of COVID-19 [2]. Africa had approximately 11.5% of all internet users globally in 2020, an increase from 10.9% in 2019 (pre-pandemic) [3, 4]. Increased internet use was highest among individuals aged 17 to 24 years (the typical age of undergraduate medical university students) [5, 6]. The increase in internet use among university students was attributed to the move from offline (in-person) teaching to online classes, where students accessed online teaching materials leading to an increased amount of time spent on the internet [1]. This increased time spent on internet increased students involvement in other internet digital platforms and electronic activities. In addition, digital platforms provided social connections and interactions, education and virtual learning, the performance of religious activities and spiritual support, and e-business which provided social, emotional, and economic aid to individuals during the height of the COVID-19 pandemic restrictions [7]. The increased use of the internet also made it easy for students to access mental health services such as online therapy sessions in a period of heightened stress, anxiety, and depression [8, 9]. These positive benefits of online internet use on well-being, quality relationships, and social connectedness have been described by Internet-enhanced self-disclosure hypothesis used by other researchers [10, 11].

However, increased internet use among adolescents and emerging adults has also been associated with online-related behavioral dependency (e.g., smartphone addiction, internet use addiction, social media addiction, online gaming addiction, etc.) which has been termed as 'problematic internet use' (PIU). PIU is the use of internet that results in psychological and social difficulties in a person's life (e.g., compromising of relationships, occupation and/or education) [12]. Problematic internet behavior is associated with a higher risk of experiencing mental health problems and psychological consequences such as depression, anxiety, sleep problems, chronic stress, and poor self-esteem [1, 13–15]. Furthermore, PIU has been found

to exacerbate negative family effects for both adolescents and their parents. For instance, studies have reported strained parent-adolescent relationships, which can negatively impact the mental health of both parties and increase their risk of depression [16]. Similarly, poor attachment of young individuals to their parents and peers has been associated with significant psychopathology, which can lead to social media addiction and various mental health issues [17].

Studies show that unpredictable psychological rewards from smartphone use in socialization, and gaming more generally, typically activate neurobiological dopaminergic pathways that increase reward-seeking behavior in anticipation of social rewards from other internet users, which in a minority of individuals can lead to dependency and addiction [18]. Moreover, COVID-19 related mental health problems have been shown to be associated with PIU. Studies have also shown that psychological distress among emerging adults appeared to result in internet addiction and Instagram addiction during the pandemic [19].

The negative effect of internet use on social relationships have also been explained by the evolutionary mismatch model by Sbarra et al. [20]. In this model, internet-related activities (smartphones use and social media) may potentially have reversed what was ancentrally an adoptive behavior regarding social bonding through self-disclosure and responsiveness into a new maladaptive behavior with superficial responses and disclosure (i.e., lack of face-to-face interaction) that is associated with mental health problems [11, 20]. The negative impacts have also been reported among other forms of internet use such as online gaming. Case study evidence has indicated that excessive online gaming was associated with extreme consequences, such as suicide among a very small minority of students during the COVID-19 pandemic [21].

However, many studies reported mixed findings regarding the association between internet use and the mental health of many young adults (such as university students) in sub-Saharan Africa, where both positive and negative impacts have been equally emphasized. For example, depression, social isolation, and the loss of loved ones to COVID-19 may have lead to increased internet dependence and addiction as a coping mechanism. Additionally, increased internet use among students may have resulted from increased academic requirements involving online classes, which may also be some of the confounding factors exacerbating increased internet use (which in extreme cases could lead to online dependency and addiction. It should be noted that most previous studies are cross-sectional, and causality between these variables cannot be determined [22].

It should be noted that not all individuals who frequently use the internet may develop problematic internet use which at its most extreme has been described as internet addiction [23]. Personality differences and characteristics have been reported to play an important role on the development and maintenance of internet addiction by several theories. For example uses and gratifications theory where different personality traits lead to different internet use motives and different types of addiction or different motivations within a specific type of problematic internet use [23]. Also, individual characteristics such as gender, level of education, age, and internet-use expectancies predict online communication applications use disorders such as social media addiction [24]. Ths has been supported by Interaction of Person-Affect-Cognition-Execution (I-PACE) model for specific internet-use disorders. This posits that individuals' addiction to specific internet applications or sites can be explained with a process that is the consequence of interactions between individuals' sociodemographic/core characteristics, different predisposing factors, mediators, and moderators [25, 26].

In Africa, internet addiction has been characterized by emotional dependence and is associated with mental health disorders such as depression [27]. Some research has indicated that students in Uganda spend considerably more hours on the internet and have higher prevalence rates of internet addiction than students in other African countries such as Namibia [27]. Use of social media has been found to be associated with lower academic performance among

university students in Uganda [28–30]. Many studies in Uganda have assessed the impact of social media use on student wellness and academic performance. However, no study has investigated the effect of smartphone addiction, social media addiction, and internet addiction simultaneously among university students in Uganda. Medical students in Uganda have been reported to have high prior knowledge and skills in internet use and commonly use social media platforms [6]. This puts medical students at higher risk of excessive internet use, which may result in detrimental psychological consequences related to internet use, such as depression. Various studies have reported a relationship between depression and internet use disorders during the pandemic [1, 13, 31, 32]. However, no previous studies have explored this relationship in Uganda despite the high prevalence of depression among university students during the COVID-19 pandemic (between 20% to 81%) [33, 34].

During the pandemic, the following factors have also been associated with depression increase including the use of psychoactive substances, having trouble paying university tuition fees, family history of mental illness, insecurity at places of residence, financial problems, romantic relationship problems, history of sexual abuse, worry about academic performance, poor sleep quality, advance childhood experiences, caring for loved ones with COVID or other medical conditions, and having a medical illness [33–38]. However, studies among medical students have also shown that the nature of the curriculum, traumatic events during hospital practice, inconsistent academic grades, and reduced free time for leisure activities are associated with a higher risk of experiencing depression [39, 40]. Furthermore, many studies have reported gender differences in predisposition to depression among medical university students (i.e., females have been reported to be at higher risk of developing depression than males [41]).

The Ugandan government first tackled the COVID-19 pandemic in March 2020, when the Ministry of Health instructed the complete closure of schools, institutions, and universities to control the spread of COVID-19 infection [42]. Most of the school system remained completely closed for two years. However, Ugandan universities transitioned to eLearning through the University Learning Management System for schooling and interactions with the students, which resulted in increased time spent on the internet among students. In Uganda, medical students were the first to be initiated into the online learning system during the COVID-19 pandemic. Therefore, they spent most of their time online compared to before the pandemic [43, 44]. Consequently, the present study explored how technology-related addictions associated with depression symptoms among undergraduate medical students in Uganda. Given that the study was an exploratory study, there were no specific hypotheses.

## Methods

### Study design and area

This was an online cross-sectional pilot survey conducted using *Google Forms*, among undergraduate medical students at the Mbarara University of Science and Technology (MUST) from November 2021 to January 2022. MUST is a public university located in southwestern Uganda with in Mbarara city. The university has 5 faculties (science, medicine, computing and informatics, business and management sciences, applied sciences and technology, interdisciplinary studies). Students in the Faculty of medicine, especially medical students, were allowed to continue studying online while others were still in complete lockdown.

### Sample size

At the time of data collection, MUST had a total of 442 undergraduate medical students in the academic year 2020/2021 (figures provided by the university administration). The minimum sample size for the present study was calculated using the Kish-Leslie formula for prevalence

studies [45], where **N** is the number of respondents needed, **p** is the estimated prevalence of depression among medical students at Makerere University in a recent study in Uganda (21.5%) [46], **Z** is 1.96 (the Z score corresponding to 95% confidence interval), and **d** is the maximum error the researcher is willing to allow (0.05).

$$N = \frac{Z^2 p(1-p)}{d^2}$$

The minimum final sample size was calculated to be 263 medical students.

## Data collection procedure and study measures

All university's social media platforms and institutional email accounts were used to recruit participants for data collection. For easy communication in the Faculty of Medicine, all undergraduate students have institutional email account and are added to an email group that acts as a communication platform between the administration and the student body. The research team distributed the survey link through these email channels and students' social media platforms such as *WhatsApp*, *Facebook*, and *Telegram* groups. In addition, some class representatives also aided in distributing the online survey link amongst their respective class members.

The online survey tool included a consent form, where informed consent was first obtained before accessing the survey. The survey consisted of sections capturing information on (i) socio-demographic information, (ii) behavioral lifestyle, (iii) online use behaviors and technological addictions (smartphone addiction, social media addiction, and internet addiction), (iv) Life stressors experienced by students (over the past two weeks), and (v) depression (using the nine-item Patient Health Questionnaire). All data were collected in English language given that all the medical students in the present study were proficient in English because all their medical training is conducted in this language. Information collected under these sections is described below:

**Socio-demographic information.** Information collected included the age (in completed years), gender (male, female), year of study, and relationship status (in a relationship, not in a relationship).

**Behavioral lifestyle.** Using a binary response (yes/no), information was collected on whether participants smoked cigarettes, smoked marijuana, were living with any chronic disease/condition, and whether they engaged in physical activities like walking, cycling, swimming, or other activities for at least 30 minutes daily. Participants were also asked the average number of hours of sleep they had per day.

**Online use behaviors and technological addictions.** These were determined by the following: online use behavior. Smartphone, social media, and internet addiction.

*Online use behaviors.* Participants were asked the following; the purpose of their internet usage (education, entertainment, or both), average number of hours spent on the internet daily, and how they accessed broadband internet access (i.e., using free university Wi-Fi, or use both mobile data and free university Wi-Fi). In addition, students were asked to select the social media platforms they currently used among the following: *Facebook*, *YouTube*, *WhatsApp*, *Instagram*, *Tiktok*, *Twitter*, *Telegram*, *Snapchat*, *Pinterest*, *LinkedIn*, *others (included Likee*, *Reddit*, *Tumblr*, and *WeChat*). These had very few users, hence were matted together.

*Smartphone addiction.* The Smartphone Application-Based Addiction Scale (SABAS) was used to assess the risk of smartphone addiction [47]. The scale has six items (e.g., *"My smartphone is the most important thing in my life"*) whose responses are rated on a six-point Likert type scale from 1 (*strongly disagree*) to 6 (*strongly agree*), and the total score ranges from 6 to 36 [47]. A higher score SABAS indicates greater risk of addiction to a smartphone application

in the individual. As reported in previous studies, individuals with a score of 21 and above were classed as being at risk of smartphone addiction [13]. In the present study, Cronbach's alpha for this tool was 0.80.

*Social media addiction.* The Bergen Social Media Addiction Scale (BSMAS) [48] was used to assess the risk of social media addiction. The scale has six items adapted from the Bergen Facebook Addiction Scale [49] (e.g., *"Used social media so much that it has had a negative impact on your job/studies?"*) whose responses are rated on a five-point Likert type scale from 1 (*very rarely*) to 5 (*very often*). The scale's total score ranges from 6 to 30, and a higher score indicates a greater social media addiction risk. As reported in previous studies, individuals with a score of 19 and above were classed as being at risk of social media addiction [50, 51]. In the present study, Cronbach's alpha was 0.87.

*Internet addiction.* The Internet Addiction Test (IAT) was used to assess the risk of internet addiction [52]. The scale has 20 items following the addition of 12 items from the original eight items [53]. The items (e.g., *"Do you fear that life without the internet would be boring, empty and joyless?"*) are rated on a six-point scale from 0 (*not applicable*) to 5 (*always*) and the total score ranges from 20 to 100. A higher score indicates a greater risk of internet addiction and a cutoff of 80 is used for the risk of internet addiction since low cutoffs inflate the risk of internet addiction. In this study, the total IAT score was used to assess for its relationship with depression. The scale has been previously validated for use among university students in Uganda [27, 54]. In the present study, Cronbach's alpha was 0.88.

**Life stressors.** Using a binary response (yes/no), participants were asked to report if they experienced any of the following problems in the past two weeks prior to the study: tuition fees payment problems, personal finance problems, romantic relationship problems, academic curriculum problems, problems relating to lack of free time, traumatic patient events (e.g., witnessing a dying patient), academic performance problems, recent acute illness problems (e.g., poor control of the condition), death of a parent/relative/friend, and having an acute physical illness (e.g., COVID-19, malaria, etc.).

**Depression.** The Patient Health Questionnaire (PHQ-9) was used to assess depression symptoms over the past two weeks. It consists of nine items (e.g., *"Feeling down, depressed, or hopeless"*) whose response is rated on a four-point scale ranging from 0 (*not at all*) to 3 (*nearly every day*). Total scores are obtained by summing the participants' responses, ranging from 0 to 27. Depression was categorized based on the following scores: 1–4 for minimal, 5–9 for mild, 10–14 for moderate, 15–19 for moderately severe, and 20–27 for severe depression [55–57]. The questionnaire is commonly used Uganda and the scale has good psychometric properties in Uganda [58, 59]. In the present study, Cronbach's alpha was 0.88.

## Ethical considerations

The present study was conducted in accordance with the Declaration of Helsinki 2013 [60] and was approved by the Mbarara University of Science and Technology research ethics committee (reference number: MUST-2021-204:Â). Furthermore, the Dean of Students at the MUST and Dean of Faculty of Medicine gave permission for data collection from students. Participation in the study was voluntary, and the survey included a detailed consent form that informed all participants about the risks and benefits of participation. Those who provided their informed consent to participate were automatically allowed entry to the study survey. Data confidentiality and anonymity were emphasized. Participants did not have to respond to all survey questions, and were free to end the survey at any point with no penalty whatsoever.

## Statistical analysis

Following data cleaning in *Microsoft Excel 16*, data were imported into STATA version 16.0 for formal data analysis. Means and standard errors were used to summarise continuous variables with normal distribution, while percentages and frequencies summarised categorical variables. Student *t*-tests and analysis of variance (ANOVA) tests were performed to identify statistically significant differences between depression symptom score and independent categorical study variables. Pearson's correlation coefficient was used to determine the relationships between continuous independent variables and depression symptoms score. Hierarchical linear regression was used to determine the predictors of depression symptoms, and five models were generated. Collinearity was tested for, and those with a variance inflation factor (VIF) of less than three were included in the model. The factors associated with depression were obtained from the final model. A $p<0.05$ for the significance level was considered at a 95% confidence interval.

# Results

## Participants' sociodemographics and behavioral lifestyles

Among the 269 participants recruited in the study, their mean age was 23.37±3.38 years, with over half of the participants being male (58.36%). Approximately one-quarter were third-year medical students (24.91%) and majority of the participants were not in a relationship (94.42%). The majority of the participants did not report any chronic illnesses (93.31%). Only six participants smoked cigarettes (2.23%), and 12 participants smoked marijuana (4.46%). Approximately three-quarters reported participation in any daily physical exercise (73.23%), and the most common social media platform used was *WhatApp* (98.51%) followed by *YouTube* (85.82%) (Table 1).

## Online-related addictions

The prevalence of being at risk of (i) smartphone addiction was 45.72%, (ii) social media addiction was 74.34%, and (iii) problematic internet use was 36.43%. The prevalence of internet addiction was 73.98% (i.e., 26.02, 32.71, and 15.24, for low level, moderate, and severe internet addiction, respectively). A total of 23 (8.55%) had high risk of internet addiction.

## Depression

Using a cutoff score of 10 on the PHQ-9, 16.73% of the medical students were classed as having moderate to severe depression symptoms. The prevalence of minimal, mild, moderate, moderately severe, and severe depression symptoms was 35.32%, 25.65%, 10.78%, 3.72%, and 2.23%, respectively. Only 60 students did not have any symptoms of depression (Fig 1).

## Relationship between depression symptom severity and study variables

**Sociodemographics and behavioral lifestyle.** Table 1 shows the relationship between independent variables and the depression symptom score. There was also a statistically significant difference between depression symptom score and gender, with the average symptoms of depression being higher among females than males (5.93±0.52 vs. 4.13±0.37, t = 2.85, $p = 0.005$). The average depression symptoms score was lowest among fifth-year medical students compared to other medical students (2.43 ± 0.60 for year five vs. 6.52 ± 0.89, 5.15 ± 0.63, 6.57 ± 0.71, 3.38 ± 0.51, for year 1, 2, 3, 4; respectively).

**Online use behaviors.** On average, most students were buying their own mobile data (Table 1). Most participants (16.8%) used five social media platforms (Fig 2).

**Table 1. Distribution of study variables in relation to severity of depression symptoms (kurtosis = 4.18) and (skewness = 1.26).**

| Variables | n (%) | Depression symptom severity | $F/t^2$ | p-value |
|---|---|---|---|---|
| | | μ ± SE | | |
| *Socio-demographic variables* | | | | |
| **Gender** | | | | |
| Female | 112 (41.64) | 5.93 ± 0.52 | 2.85 | **0.005** |
| Male | 157 (58.36) | 4.13 ± 0.37 | | |
| **Year of study** | | | | |
| First | 40 (14.87) | 6.52 ± 0.89 | 7.17 | **<0.001** |
| Second | 60 (22.30) | 5.15 ± 0.63 | | |
| Third | 67 (24.91) | 6.57 ± 0.71 | | |
| Fourth | 58 (21.56) | 3.38 ± 0.51 | | |
| Fifth | 44 (16.36) | 2.43 ± 0.60 | | |
| **Relationship status** | | | | |
| In a relationship | 15 (5.58) | 3.33 ± 0.96 | 1.43 | 0.232 |
| Not in a relationship | 254 (94.42) | 4.97 ± 0.33 | | |
| *Behavioral lifestyle variables* | | | | |
| **Smoked cigarettes** | | | | |
| No | 263 (97.77) | 4.83 ± 0.31 | -1.02 | 0.309 |
| Yes | 6 (2.23) | 7.00 ± 2.96 | | |
| **Smoked marijuana** | | | | |
| No | 257 (95.54) | 4.81 ± 0.32 | -1.00 | 0.319 |
| Yes | 12 (4.46) | 6.33 ± 1.61 | | |
| **Engaged in daily physical exercise** | | | | |
| No | 72 (26.77) | 5.18 ± 0.59 | 0.57 | 0.565 |
| Yes | 197 (73.23) | 4.77 ± 0.37 | | |
| **Had a chronic illness** | | | | |
| No | 251 (93.31) | 4.89 ± 0.32 | 0.09 | 0.930 |
| Yes | 18 (6.69) | 4.78 ± 1.47 | | |
| *Online use behavior variables* | | | | |
| **Purpose of internet usage** | | | | |
| Education | 13 (4.83) | 4.46 ± 1.38 | 1.42 | 0.243 |
| Entertainment | 8 (2.97) | 7.88 ± 1.56 | | |
| Both | 248 (92.19) | 4.80 ± 0.33 | | |
| **Methods of internet access broadband** | | | | |
| Both Mobile data and University Wi-Fi | 62 (23.05) | 5.43 ± 0.75 | 0.93 | 0.335 |
| Mobile data | 207 (76.95) | 4.71 ± 0.34 | | |
| **Social media platforms used** | | | | |
| *Facebook* | | | | |
| No | 121 (45.15) | 4.77 ± 0.46 | -0.35 | 0.725 |
| Yes | 147 (54.85) | 5.00 ± 0.42 | | |
| *YouTube* | | | | |
| No | 38 (14.18) | 4.10 ± 0.76 | -1.02 | 0.306 |
| Yes | 230 (85.82) | 5.03 ± 0.34 | | |
| *WhatsApp* | | | | |
| No | 4 (1.49) | 3.00 ± 2.12 | -0.74 | 0.459 |
| Yes | 264 (98.51) | 4.93 ± 0.32 | | |
| *Instagram* | | | | |

*(Continued)*

**Table 1.** (Continued)

| Variables | n (%) | Depression symptom severity | F/t² | p-value |
|---|---|---|---|---|
| | | μ ± SE | | |
| No | 137 (51.12) | 4.69 ± 0.46 | -0.69 | 0.490 |
| Yes | 131 (48.88) | 5.12 ± 0.43 | | |
| *TikTok* | | | | |
| No | 174 (64.93) | 4.14 ± 0.37 | -3.32 | **0.001** |
| Yes | 94 (35.07) | 6.29 ± 0.55 | | |
| *Twitter* | | | | |
| No | 109 (40.67) | 5.46 ± 0.53 | 1.50 | 0.135 |
| Yes | 159 (59.33) | 4.51 ± 0.38 | | |
| *Telegram* | | | | |
| No | 120 (44.78) | 4.53 ± 0.47 | -1.05 | 0.296 |
| Yes | 148 (55.22) | 5.20 ± 0.42 | | |
| *Snapchat* | | | | |
| No | 190 (70.90) | 4.19 ± 0.34 | -3.59 | **<0.001** |
| Yes | 78 (29.10) | 6.63 ± 0.63 | | |
| *Pinterest* | | | | |
| No | 227 (84.70) | 4.75 ± 0.33 | -1.12 | 0.262 |
| Yes | 41 (15.30) | 5.73 ± 0.87 | | |
| *LinkedIn* | | | | |
| No | 224 (83.58) | 4.88 ± 0.34 | -0.11 | 0.912 |
| Yes | 44 (16.42) | 4.98 ± 0.86 | | |
| *Others*[#] | | | | |
| No | 249 (92.57) | 4.79 ± 0.32 | 1.11 | 0.293 |
| Yes | 20 (7.46) | 6.05 ± 1.47 | | |
| *Technology addictions* | | | | |
| **Smartphone addiction** | | | | |
| Normal | 146 (54.28) | 2.76 ± 0.30 | -8.18 | **<0.001** |
| Risk of addiction | 123 (45.72) | 7.39 ± 0.50 | | |
| **Social media addiction** | | | | |
| Normal | 62 (23.05) | 1.69 ± 0.28 | -5.88 | **<0.001** |
| Risk of addiction | 200 (76.95) | 5.83 ± 0.37 | | |
| **Internet addiction** | | | | |
| **Internet addiction (cut off score of 80)** | | | | |
| No | 246 (91.45) | 4.42 ± 0.31 | -5.02 | **<0.001** |
| Yes | 23 (8.55) | 1.31 ± 6.28 | | |
| *Life stressors experienced over the past two weeks* | | | | |
| **Tuition fee payment problems** | | | | |
| No | 210 (78.36) | 4.55 ± 0.35 | -2.11 | **0.036** |
| Yes | 58 (21.64) | 6.16 ± 0.66 | | |
| **Personal financial problems** | | | | |
| No | 96 (35.82) | 3.46 ± 0.45 | -3.488 | **<0.001** |
| Yes | 172 (64.18) | 5.70 ± 0.41 | | |
| **Romantic relationship problems** | | | | |
| No | 167 (62.31) | 3.16 ± 0.32 | -7.89 | **<0.001** |
| Yes | 101 (37.69) | 7.78 ± 0.54 | | |
| **Academic curriculum problems (lectures, ward round, tests, examination)** | | | | |

*(Continued)*

**Table 1.** (Continued)

| Variables | n (%) | Depression symptom severity | $F/t^2$ | *p*-value |
|---|---|---|---|---|
| | | μ ± SE | | |
| No | 135 (50.37) | 3.25 ± 0.36 | -5.56 | **<0.001** |
| Yes | 133 (49.63) | 6.57 ± 0.48 | | |
| **Problems related to lack of free time** | | | | |
| No | 145 (54.10) | 3.35 ± 0.34 | -5.63 | **<0.001** |
| Yes | 123 (45.90) | 6.72 ± 0.21 | | |
| **Witnessing a traumatic event in the hospital (e.g., patient death)** | | | | |
| No | 213 (79.48) | 4.72 ± 0.34 | -1.10 | 0.272 |
| Yes | 55 (20.55) | 5.58 ± 0.73 | | |
| **Academic performance problems** | | | | |
| No | 144 (53.73) | 2.93 ± 0.31 | -7.38 | **<0.001** |
| Yes | 124 (46.27) | 7.19 ± 0.51 | | |
| **Recent acute illness problems (e.g., poor control of the illness)** | | | | |
| No | 247 (92.16) | 4.67 ± 0.32 | -2.54 | **0.012** |
| Yes | 21 (7.84) | 7.62 ± 1.35 | | |
| **Death of a parent/relative/friend** | | | | |
| No | 218 (81.34) | 4.40 ± 0.33 | -3.38 | **<0.001** |
| Yes | 50 (18.66) | 7.08 ± 0.81 | | |
| **Having an acute physical illness (non-chronic such as malaria, flu, COVID-19)** | | | | |
| No | 235 (87.69) | 4.59 ± 0.32 | -2.64 | **0.009** |
| Yes | 33 (12.31) | 7.09 ± 1.07 | | |

Others [#]: included Reddit, WeChat, Tumblr, Likee.

**Technological addictions.** On average depression was significantly more among participants at risk of smartphone addiction (*t* = -8.18; *p*<0.001), social media addiction (*t* = -5.88; *p*<0.001), and internet addiction (*t* = -5.02, *p* = 0.001). For details see Table 1.

**Life stressors over past two weeks.** On average individuals who reported to have experienced life stressors bothering them had more depression symptoms. However, there was statistically significant differences with almost all the explored life stressors apart from recent acute illness problems (e.g., poor control of the illness). For details see Table 1.

## Correlations between continuous variables and depression symptoms

Table 2 reports the different correlations between continuous independent variables and depression symptoms. Statistically significant moderate positive correlations were found between depression and risk of social media addiction ($r^2 = 0.51$), and internet addiction severity ($r^2 = 0.57$). A lower positive significant correlation was found between depression and risk of smartphone addiction ($r^2 = 0.49$). Negligible statistically significant correlations were found between depression with age ($r^2 = -0.20$) and average hours spent daily on the internet ($r^2 = 0.23$). High positive statistically significant correlations were found between the following variables: (i) risk of social media addiction and risk of smartphone addiction ($r^2 = 0.74$), and (ii) risk of internet addiction and risk of social media addiction ($r^2 = 0.72$).

## Prediction models for depression symptom severity

Utilizing hierarchical linear regression modeling, five models were tested to predict participant depression symptom severity. Multicollinearity was tested for the variables included in each

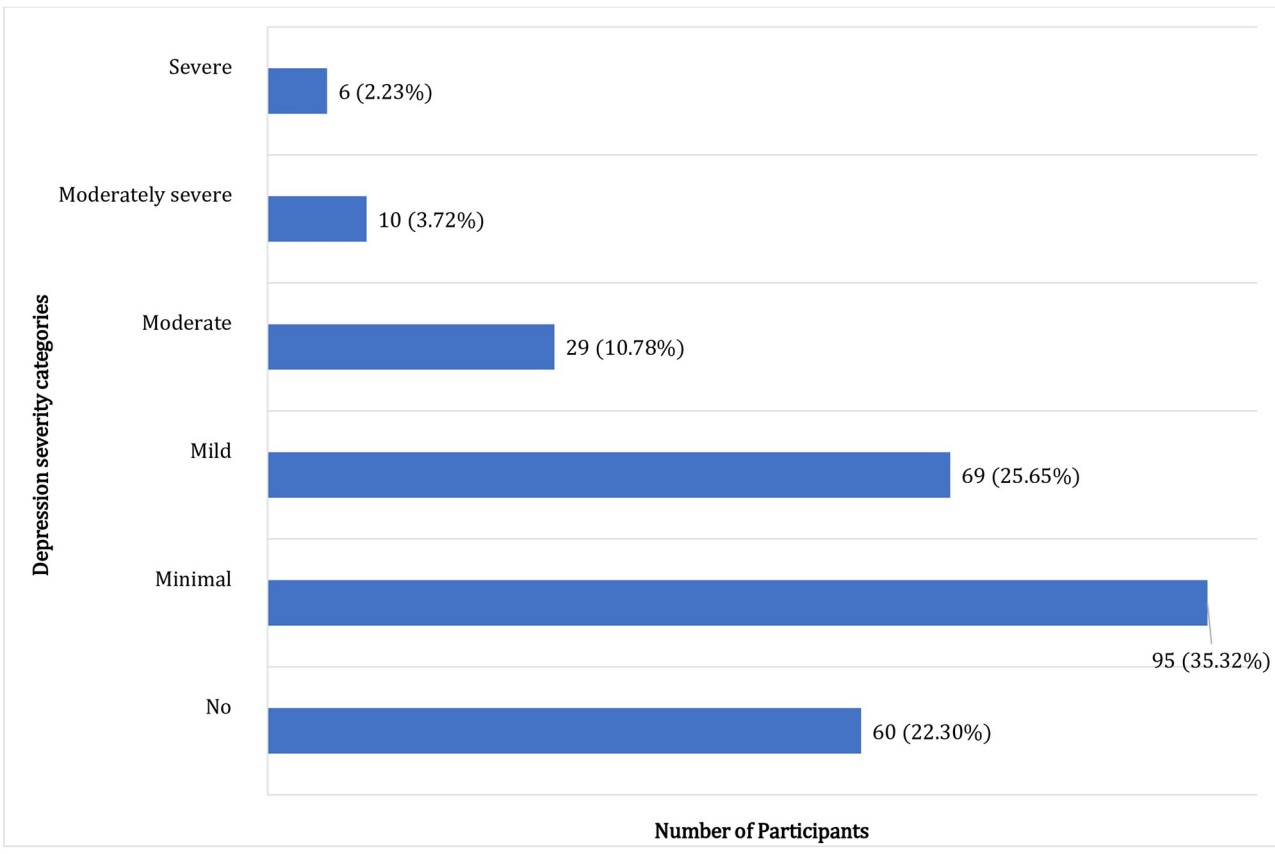

**Fig 1. Depression severity categories based on PHQ-9 scores among the study participants.**

model and each variable exhibited a VIF less than 3. Model 1 had sociodemographics and a history of chronic illness, which predicted 8.83% of the depression symptom severity among the participants. When participants' behavioral lifestyle variables were added in Model 2, the cumulative variance in depression score severity increased slightly by 6%. On addition of the life stressors reported in the past two weeks to Model 3, the cumulative variance in depression score severity increased to 35.9%, making lifestyle stressors the largest predictor of depression compared to other predictor variables. Online use behaviors and participants' online addiction scores were added to Models 4 and 5, respectively, increasing the variance of depression symptom severity to 41.4% and 51.9% respectively (Table 3). In the final model, the factors associated with an increase in severity of depression symptoms were experiencing romantic relationship problems ($\beta$ = 2.30, 95% standard error ($SE$) = 0.58, $p<0.001$), academic performance problems ($\beta$ = 1.76, $SE$ = 0.60, $p<0.001$), and internet addiction severity score ($\beta$ = 0.05, $SE$ = 0.02, $p<0.001$). however, using the *Twitter* social media platform was found to reduce the depression symptom severity score ($\beta$ = -1.88, $SE$ = 0.57, $p<0.05$).

## Discussion

The present study is the first of its kind in Uganda to be conducted among medical students to explore the association between the internet use behaviors on depression. The findings indicated that 16.73% of medical students had depression, and its severity was majorly positively associated with internet addiction severity scores. The prevalence of depression reported in

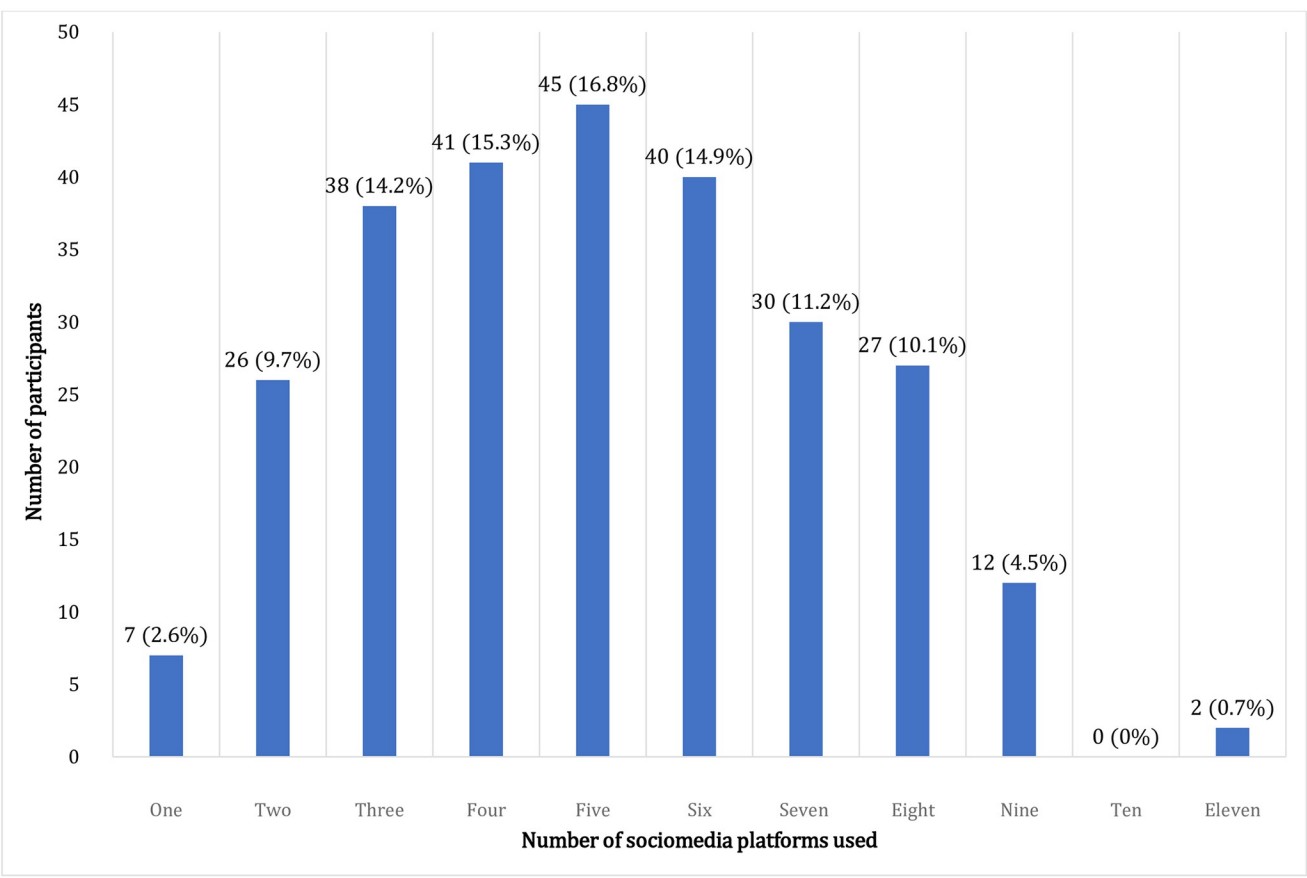

**Fig 2. Number of social media platforms used.**

Table 2. Correlations between continuous variables and depression symptoms score.

| Variable | (μ±SE) | 1 | 2 | 3 | 4 | 5 | 6 | 7 | 8 |
|---|---|---|---|---|---|---|---|---|---|
| Age (1) | 23.37 ± 3.38 | 1 | | | | | | | |
| Hours of daily sleep (2) | 6.99±2.54 | -0.06 | 1 | | | | | | |
| Average hours spent on internet (3) | 6.99±4.41 | **-0.12**\* | 0.11 | 1 | | | | | |
| Number of social media platforms used (4) | 5.07±2.12 | -0.15 | -0.05 | **0.16**\* | 1 | | | | |
| Smartphone addiction score (5) | 19.72±6.63 | -0.02 | 0.02 | **0.24**\*\* | **0.31**\* | 1 | | | |
| Social media addiction score (6) | 24.50±9.13 | -0.04 | 0.02 | **0.22**\*\* | **0.19**\* | **0.74**\*\* | 1 | | |
| Internet addiction score (7) | 34.40±25.83 | -0.09 | 0.02 | **0.26**\*\* | **0.21**\* | **0.67**\*\* | **0.72**\*\* | 1 | |
| Depression severity score (8) | 4.88±5.15 | **-0.20**\*\* | 0.01 | **0.23**\*\* | 0.13 | **0.49**\*\* | **0.51**\*\* | **0.57**\*\* | 1 |

\* = $p$-value less than 0.05;

\*\* = $p$-value less than 0.01;

very high correlation positive (negative) = $r^2$ = 0.90 to 1.00 (−0.90 to −1.00); high positive (negative) correlation = 0.70 to 0.90 (−0.70 to −0.90); moderate positive (negative) correlation = 0.50 to 0.70 (−0.50 to −0.70); low positive (negative) correlation = 0.30 to 0.50 (−.30 to −0.50); Negligible correlation = 0.00 to 0.30 (.00 to −0.30).

**Table 3. Predictive model for depression symptom severity.**

| Variables | | Model 1 | Model 2 | Model 3 | Model 4 | Model 5 |
|---|---|---|---|---|---|---|
| | | $F = 5.10$, $p < 0.001$, $r^2 = 0.0883$ | $F = 2.97$, $p = 0.002$, $r^2 = 0.0936$ | $F = 7.30$, $p < 0.001$, $r^2 = 0.3588$ | $F = 5.01$, $p < 0.001$, $r^2 = 0.4139$ | $F = 6.93$, $p < 0.001$, $r^2 = 0.5193$ |
| | | $\beta$ (SE) | $\beta$ (SE) | $\beta$ (SE) | $\beta$ (SE) | $\beta$ (SE) |
| Constant | | 12.50 (4.98) * | 11.22 (5.63) * | 8.65 (5.11) | 6.62 (5.68) | 0.79 (5.25) |
| **Sociodemographic and history of chronic illness** | Age | -0.14 (0.13) | -0.14 (0.14) | -0.21 (0.12) | -0.17 (0.13) | -0.15 (0.12) |
| | Gender | -1.18 (0.64) | -1.19 (0.66) | -1.29 (0.58) * | -0.85 (0.65) | -0.71 (0.59) |
| | Year of study | -0.79 (0.26) * | -0.79 (0.27) * | -0.17 (0.24) | -0.18 (0.24) | 0.05 (0.22) |
| | Relationship status | 0.10 (0.89) | 0.12 (0.91) | -0.23 (0.80) | 0.18 (0.83) | 0.46 (0.76) |
| | Chronic illness | -0.38 (1.22) | -0.35 (1.23) | -0.20 (1.31) | -0.35 (1.36) | -0.70 (1.25) |
| **Behavioral lifestyle variables** | Smoked cigarettes | | 0.77 (2.58) | 1.31 (2.23) | 1.26 (2.25) | 0.51 (2.07) |
| | Smoked marijuana | | 1.27 (1.89) | 0.24 (1.64) | 0.13 (1.67) | 1.14 (1.55) |
| | Engaged in daily physical exercise | | -0.27 (0.72) | 0.06 (0.63) | -0.26 (0.64) | 0.12 (0.59) |
| | Hours of sleep daily | | -0.07 (0.12) | -0.07 (0.11) | -0.11 (0.11) | -0.15 (0.10) |
| **Life stressors (over the past two weeks)** | Problems paying university tuition fees | | | 0.15 (0.71) | -0.26 (0.74) | -0.44 (0.68) |
| | Personal financial problems | | | 0.54 (0.62) | 0.76 (0.63) | 0.71 (0.57) |
| | Romantic relationship problems | | | 3.10 (0.61) ** | 3.09 (0.62) ** | **2.30 (0.58) ** |
| | Academic curriculum problems (lectures, ward round, tests, examination) | | | -0.20 (0.68) | -0.08 (0.70) | -0.30 (0.64) |
| | Problems relating to lack of free time | | | 1.67 (0.62) * | 1.24 (0.64) | 0.36 (0.60) |
| | Witnessing a traumatic event in the hospital (e.g., patient death) | | | -1.05 (0.70) | -1.04 (0.70) | -0.87 (0.64) |
| | Academic performance problems | | | 2.34 (0.64) ** | 2.13 (0.66) ** | **1.76 (0.60) ** |
| | Recent acute illness problems (e.g., poor control of the illness) | | | 0.73 (1.42) | 0.32 (1.44) | -0.13 (1.32) |
| | Death of a parent/relative/friend | | | 1.28 (0.71) | 1.38 (0.71) | 0.91 (0.65) |
| | Having an acute physical illness (e.g., malaria, flu, COVID-19, etc.) | | | 0.14 (1.09) | 0.36 (1.10) | 0.23 (1.00) |
| **Online use behaviours** | Purpose of internet usage | | | | 0.79 (0.69) | 0.73 (0.63) |
| | Average hours spent daily on internet | | | | 0.16 (0.07) * | 0.08 (0.61) |
| | Methods of internet access broadband | | | | -0.69 (0.68) | -0.29 (0.62) |
| | Facebook | | | | 0.13 (0.61) | -0.32 (0.56) |
| | YouTube | | | | 0.54 (0.80) | 0.29 (0.74) |
| | WhatsApp | | | | -1.11 (2.26) | 0.57 (2.09) |
| | Instagram | | | | -0.08 (0.63) | 0.36 (0.58) |
| | TikTok | | | | 0.86 (0.66) | 0.31 (0.60) |
| | Twitter | | | | -1.72 (0.62) * | **-1.88 (0.57) ** |
| | Telegram | | | | -0.15 (0.63) | -0.26 (0.58) |
| | Snapchat | | | | 0.38 (0.78) | 0.09 (0.72) |
| | Pinterest | | | | -1.08 (0.86) | -0.57 (0.79) |
| | LinkedIn | | | | 0.66 (0.77) | 0.12 (0.71) |
| | Others # | | | | 1.50 (1.10) | 1.07 (1.01) |

(*Continued*)

**Table 3.** (Continued)

| Variables | | Model 1 | Model 2 | Model 3 | Model 4 | Model 5 |
|---|---|---|---|---|---|---|
| | | $F = 5.10$, $p<0.001$, $r^2 = 0.0883$ | $F = 2.97$, $p = 0.002$, $r^2 = 0.0936$ | $F = 7.30$, $p<0.001$, $r^2 = 0.3588$ | $F = 5.01$, $p<0.001$, $r^2 = 0.4139$ | $F = 6.93$, $p<0.001$, $r^2 = 0.5193$ |
| | | $\beta$ (SE) | $\beta$ (SE) | $\beta$ (SE) | $\beta$ (SE) | $\beta$ (SE) |
| Technology-related addictions | Smart phone addiction | | | | | 0.07 (0.06) |
| | Social media addiction | | | | | 0.05 (0.04) |
| | Internet addiction | | | | | **0.05 (0.02)**\*\* |

Others [#]: included Reddit, WeChat, Tumblr, Likee

the present study (16.73%) is lower than another study in Uganda assessing depression using the same instrument at a different university in 2019 (21.5%) [46]. These differences may be because of the different curricula, teaching methods, and mental wellness programs exhibited by the 2 institutions. There is also a possibility of social-cultural differences because the Makerere University is located in the capital city of Uganda, Kampala, and is more urbanly located [61] compared to the university in the present study. However, the prevalence of depression was higher than the 4.0% reported among first-year medical students at Makerere University in 2002 using the Beck Depression Inventory (BDI-II) [62]. This difference may stem from the psychometric properties of the different assessment tools, where PHQ-9, being shorter and based on diagnostic criteria for depression presents an advantage and a more rigorous assessment over BDI-II [63]. In addition, there has been an increase in internet use and online-associated stressors as compared to early years of 2000s.

The prevalence rate of depression in the present study is also lower than 28% from a systematic review of 77 studies of depression among medical students globally published before April 2015 (95% confidence interval of 24.2%-32.1%) [64]. However, this review only included studies from continents other than Africa comprising undergraduate and graduate students. Graduate students have consistently been reported to have higher levels of depression [64], a factor attributed to being involved in higher intensive clinical work with an increased academic burden that leads to burnout [65]. Although the present study's prevalence rate is lower than most prevalence rates among medical students, the prevalence is higher than the national prevalence of 4.6% in Uganda [66]. This large difference shows the need to implement better mental health services to combat depression among Ugandan medical students.

The prevalence of being at risk of smartphone addiction was 45.72% in the present study, which is lower than 86.9% among Bangladeshi university students using the same instrument [13], lower than 52.7% among medical students in India using the short version of the Smartphone Addiction Scale [67], and lower than 52.8% among medical students in China using the Smartphone addiction test [68] all conducted during the first year of the COVID-19 pandemic. Despite the overall increase in smartphone use, the difference may be due geographical locations and the overall technological advancements of the different countries, where Uganda, a low income country still has poor internet bandwidth ith limited access to smartphones as compared to other countries. This study's prevalence rate of smartphone addiction was higher than in many previous studies among medical students before the COVID-19 pandemic, ranging between 22.2% and 36.5% [15, 69–73]. However, the prevalence of smartphone addiction among medical students before the pandemic in Egypt was higher (53.6% and 74.7%) than that in the present study [74, 75]. This may be because Egypt has consistently had more

smartphone users and internet users than Uganda [3, 4, 74] owing to their better internet service provisions and increased online users.

The prevalence of being at risk of social media addiction was 74.34% in the present study. This prevalence was higher than a particular social media platform use addiction (*Facebook*) among medical students in Egypt (26.9%) and Malaysia (15.5%) in 2014–2015 [76]. The difference may be due to the marked increase in social media use over the past few years, and some of the most used social media platforms, such as *WhatsApp*, *TikTok*, *Twitter*, and *YouTube*, were not assessed for in the study among Egyptian and Malaysian medical students [77].

About 8.55% of medical students in the present study screened positive for internet addiction. These were lower than 51.7% among medical students in Egypt during the first year of the COVID-19 pandemic based on the same tool and cutoff [78]. Similar to smartphone addiction, the prevalence of internet addiction in the present study was higher than many studies conducted before the COVID-19 pandemic that, ranged from 0 and 7.86 [79–85]. The difference between the pre-pandemic and during the pandemic may be due to the following reasons: (i) an increase in internet use over the years due to the world becoming more digital, and (ii) the methods used to prevent the spread of COVID-19 led to many individuals to rely on the internet, social media, and their smartphones to maintain communication and keep up with the news about the pandemic and spread of the virus.

The present study generated five step-wise models to understand depression symptom severity among medical students, i.e., beginning with sociodemographic characteristics and chronic illness presence (Model 1), and then subsequently adding behavioral life variables (Model 2), experiencing life stressors over the past two weeks (Model 3), online behaviors (Model 4), and online-related addictions (Model 5). In the past two weeks, life stressors experienced by medical students were the biggest predictor of depression severity (explaining 26% of the variance), followed by online-related addictions (explaining 10% of the variance). Behavioral lifestyle variables (psychoactive substance use [cigarettes and marijuana], hours of daily sleep, and involvement in daily physical exercise) had the least predictability and only explained 0.53 of the variance in depression severity. The final model explained 54% of the variation in depression symptom severity among medical students. With all models being statistically significant, it indicates the importance of all these factors in explaining depression symptom severity among medical students.

As reported in previous studies, depression was associated with recent romantic relationship problems [86–88]. There is a bidirectional relationship between depression and romantic relationship problems, as discussed by Vujeva and Furman [86]. In the present study, online-related addictions were predictors of depression, and the internet addiction severity score was associated with depression in the final model. Other researchers have reported a significant relationship between depression and internet addiction among undergraduate medical students [89]. This could be due to students using the internet as a coping mechanism for untreated depression, which fuels internet use behaviors and can become an addiction [89]. Dong et al. (2011) suggested that depression can be a direct consequence of internet addiction [90] because internet behaviors such as using social media are spaces where individuals can experience cyberbullying and online trolling [91].

Academic performance problems were associated with depression symptoms in the present study. This is a viscious cycle common among university students, where low academic grades, increases number of course units to retake, and high academic burden may result into feelings of anxiety and depression, which self-perpetuates into further depression and worsening academic achievemts [92]. However, individuals using the *Twitter* social platform had fewer depression symptoms than those not using *Twitter*. Symptoms of depression include a lack of motivation to read longer posts, low energy, anhedonia, and memory problems. This may

mean that 280-character posts are more suitable for those prone to depression to stay interacting with peers for social support, protecting them against depression. The reasons for the protective nature of using *Twitter* for depression require further research. However, other researchers have found *Twitter* to be associated with more depression because those experiencing depression cannot express themselves when using *Twitter* due to the low character limit [93].

### Limitations and future research

The present study has several limitations. First, a cross-sectional study design was used so that causality between the variables explored could not be determined. A longitudinal study using a larger sample size is therefore needed to understand the causal relationships between the study variables. Second, the study was conducted in one university and the sample size was small, with data collected via convenience sampling. Therefore, the findings cannot be generalized to all medical students in Uganda. Future studies examining the role of online behaviors and their impact on depression with larger and more representative samples are needed to expand the study area. Third, the PHQ-9 that was used to screen for depression has been reported to occasionally report false negative cases among the general population [94]. However, it has shown good psychometric properties in screening for depression among medical students in Uganda [33, 46, 58, 62, 65]. Fourth, despite the consequences of internet use, such as online addictions (internet, social media, and smartphone addiction) and depression, the present study did not include non-medical students. Therefore, the findings cannot be generalized to university students more generally. Future researchers should conduct a comparative study among medical and non-medical students to show which students should be given more emphasis on preventing online use consequences. Fifith, apart from the Internet Addiction Test (IAT) [27, 54], no other online addiction tool used in the present study has been validated for medical students in Uganda. Future studies should seek to validate these scales to be culturally acceptable among students in Uganda. Sixth, despite the final model explaining over half of the variance in depression symptom severity among medical students, further studies are needed to explore the relationship between depression and other factors so that possible interventions targeting the strongest predictors of depression can be designed. Seventh, the study was based on only self-report measures that might have introduced recall (and other methods) biases. Finally, some variables (such as the necessary use of the internet for academic purposes and use of the internet to keep in touch with family and friends during the pandemic) were not examined in this study, and which may have had a confounding effect on depression symptoms.

### Conclusions

Depression affects many medical students at the Mbarara University of Science and Technology and is associated with problematic online use and other acute stressors. Therefore, it is recommended that students' mental health care services consider digital wellbeing and its relationship with problematic online use as part of a more holistic depression prevention and resilience programs.

### Acknowledgments

The authors would like to thank the participants for voluntarily participating in the study. The authors would also like to thank the MUST Faculty of Medicine for allowing their students to participate in the study and assisting in recruiting the participants. The authors also appreciate

Felix Onyango Kijoiki (deceased–succumbed to COVID-19), Beneth Kirangi Tusiime, and Jannet Malaka for their contribution during data collection.

## Author Contributions

**Conceptualization:** Jonathan Sserunkuuma, Mark Mohan Kaggwa.

**Data curation:** Jonathan Sserunkuuma, Mark Mohan Kaggwa, Moses Muwanguzi, Sarah Maria Najjuka, Nathan Murungi, Jonathan Kajjimu, Jonathan Mulungi, Raymond Bernard Kihumuro.

**Formal analysis:** Mark Mohan Kaggwa, Moses Muwanguzi, Sarah Maria Najjuka.

**Supervision:** Mark Mohan Kaggwa, Scholastic Ashaba.

**Visualization:** Mark Mohan Kaggwa, Moses Muwanguzi, Mohammed A. Mamun.

**Writing – original draft:** Jonathan Sserunkuuma, Mark Mohan Kaggwa.

**Writing – review & editing:** Jonathan Sserunkuuma, Mark Mohan Kaggwa, Moses Muwanguzi, Sarah Maria Najjuka, Nathan Murungi, Jonathan Kajjimu, Jonathan Mulungi, Raymond Bernard Kihumuro, Mohammed A. Mamun, Mark D. Griffiths, Scholastic Ashaba.

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
