## [Decision Letter · Decision Letter 0]

28 Apr 2023

PONE-D-23-01046Problematic use of the internet, smartphones, and social media among medical students and relationship with depression: An exploratory studyPLOS ONE

Dear Dr. Muwanguzi,

Thank you for submitting your manuscript to PLOS ONE. After careful consideration, we feel that it has merit but does not fully meet PLOS ONE’s publication criteria as it currently stands. Therefore, we invite you to submit a revised version of the manuscript that addresses the points raised during the review process.

We look forward to receiving your revised manuscript.

Kind regards,

Giulia Ballarotto

Academic Editor

PLOS ONE

Journal Requirements:

Reviewers' comments:

Reviewer's Responses to Questions

**Comments to the Author**

1. Is the manuscript technically sound, and do the data support the conclusions?

Reviewer #1: Yes

Reviewer #3: Partly

2. Has the statistical analysis been performed appropriately and rigorously? 

Reviewer #1: Yes

Reviewer #3: Yes

3. Have the authors made all data underlying the findings in their manuscript fully available?

Reviewer #1: Yes

Reviewer #3: Yes

4. Is the manuscript presented in an intelligible fashion and written in standard English?

Reviewer #1: Yes

Reviewer #3: Yes

5. Review Comments to the Author

Reviewer #1: Kindly add a note about limitations of the study. For example, Limitations of the Patient Health Questionnaire in Identifying Anxiety and Depression. Ref Eack SM, Greeno CG, Lee BJ. Limitations of the Patient Health Questionnaire in Identifying Anxiety and Depression: Many Cases Are Undetected. Res Soc Work Pract. 2006 Nov 1;16(6):625-631. doi: 10.1177/1049731506291582. PMID: 24465121; PMCID: PMC3899353.

Reviewer #3: Thank you very much for the possibility to review the study "Problematic use of the internet, smartphones, and social media among medical students and relationship with depression: An exploratory study". The paper is interesting and well written.There are just a few points that should be further investigated. Please see my specific comments below.

1. In the Introduction the literature related to problematic Internet use should be deepened. For example, the following are suggested:

Ballarotto, G., Marzilli, E., Cerniglia, L., Cimino, S., & Tambelli, R. (2021). How does psychological distress due to the COVID-19 pandemic impact on internet addiction and Instagram addiction in emerging adults?. International journal of environmental research and public health, 18(21), 11382.

Spada, M. M. (2014). An overview of problematic Internet use. Addictive behaviors, 39(1), 3-6.

Tambelli, R., Cimino, S., Marzilli, E., Ballarotto, G., & Cerniglia, L. (2021). Late Adolescents’ Attachment to Parents and Peers and Psychological Distress Resulting from COVID-19. A Study on the Mediation Role of Alexithymia. International Journal of Environmental Research and Public Health, 18(20), 10649.

Deutrom, J., Katos, V., & Ali, R. (2022). Loneliness, life satisfaction, problematic internet use and security behaviours: re-examining the relationships when working from home during COVID-19. Behaviour & Information Technology, 41(14), 3161-3175.

Ballarotto, G., Volpi, B., & Tambelli, R. (2021). Adolescent attachment to parents and peers and the use of Instagram: The mediation role of psychopathological risk. International journal of environmental research and public health, 18(8), 3965.

Özaslan, A., Yıldırım, M., Güney, E., Güzel, H. Ş., & İşeri, E. (2022). Association between problematic internet use, quality of parent-adolescents relationship, conflicts, and mental health problems. International Journal of Mental Health and Addiction, 20(4), 2503-2519.

2. the authors used the English version of the tools? how come?

6. PLOS authors have the option to publish the peer review history of their article (what does this mean?). If published, this will include your full peer review and any attached files.

Reviewer #1: **Yes: **Dr. (Prof.) Abhishek Singh

Reviewer #3: No

---

## [Author Response · Author response to Decision Letter 0]

10 May 2023

Responses to Editors’ comments

Comment: Please ensure that your manuscript meets PLOS ONE's style requirements, including those for file naming. The PLOS ONE style templates can be found at 

Response: We have edited and renamed all files as required.

Comment: Please provide additional details regarding participant consent. In the ethics statement in the Methods and online submission information, please ensure that you have specified what type you obtained (for instance, written or verbal, and if verbal, how it was documented and witnessed). If your study included minors, state whether you obtained consent from parents or guardians. If the need for consent was waived by the ethics committee, please include this information.

Response: Consent details have been added to the revised Methods section of the manuscript.

Comment: Your ethics statement should only appear in the Methods section of your manuscript. If your ethics statement is written in any section besides the Methods, please delete it from any other section.

Response: The ethics statement under ‘Declarations’ has been removed to maintain the only one under methods section.

Comment: Please review your reference list to ensure that it is complete and correct. If you have cited papers that have been retracted, please include the rationale for doing so in the manuscript text, or remove these references and replace them with relevant current references. Any changes to the reference list should be mentioned in the rebuttal letter that accompanies your revised manuscript. If you need to cite a retracted article, indicate the article’s retracted status in the References list and also include a citation and full reference for the retraction notice.

Response: The reference list is complete and appropriately formatted. 

Responses to Reviewer 1 comments

Comment: Kindly add a note about limitations of the study. For example, Limitations of the Patient Health Questionnaire in Identifying Anxiety and Depression. Ref Eack SM, Greeno CG, Lee BJ. Limitations of the Patient Health Questionnaire in Identifying Anxiety and Depression: Many Cases Are Undetected. Res Soc Work Pract. 2006 Nov 1;16(6):625-631. doi: 10.1177/1049731506291582. PMID: 24465121; PMCID: PMC3899353. 

Response: Thanks for the suggestion. This limitation has been added to the ‘Limitations and future research’ section on page 23 lines 499–502 and states that…

“Third, the PHQ-9 that was used to screen for depression has been reported to occasionally report false negative cases among the general population [94]. However, it has shown good psychometric properties in screening for depression among medical students in Uganda [33, 46, 58, 62, 65].”

 

Responses to Reviewer 3 comments

Comment: In the Introduction the literature related to problematic Internet use should be deepened. For example, the following are suggested:

• Ballarotto, G., Marzilli, E., Cerniglia, L., Cimino, S., & Tambelli, R. (2021). How does psychological distress due to the COVID-19 pandemic impact on internet addiction and Instagram addiction in emerging adults?. International journal of environmental research and public health, 18(21), 11382.

• Spada, M. M. (2014). An overview of problematic Internet use. Addictive behaviors, 39(1), 3-6.

• Tambelli, R., Cimino, S., Marzilli, E., Ballarotto, G., & Cerniglia, L. (2021). Late Adolescents’ Attachment to Parents and Peers and Psychological Distress Resulting from COVID-19. A Study on the Mediation Role of Alexithymia. International Journal of Environmental Research and Public Health, 18(20), 10649.

• Deutrom, J., Katos, V., & Ali, R. (2022). Loneliness, life satisfaction, problematic internet use and security behaviours: re-examining the relationships when working from home during COVID-19. Behaviour & Information Technology, 41(14), 3161-3175.

• Ballarotto, G., Volpi, B., & Tambelli, R. (2021). Adolescent attachment to parents and peers and the use of Instagram: The mediation role of psychopathological risk. International journal of environmental research and public health, 18(8), 3965.

• Özaslan, A., Yıldırım, M., Güney, E., Güzel, H. Ş., & İşeri, E. (2022). Association between problematic internet use, quality of parent-adolescents relationship, conflicts, and mental health problems. International Journal of Mental Health and Addiction, 20(4), 2503-2519

Response: Thank you for your comment and for providing additional literature related to problematic internet use. We have now included a more detailed discussion of the relevant literature in the revised ‘Introduction’ section of our manuscript. We hope that the revised version increases the clarity and readability of our paper. Thank you again for your valuable feedback. 

Comment: The authors used the English version of the tools. How come?

Response: Thank you for your question. This is very simple. All our participants were medical students who are proficient in English as they conduct all their university academic activities in this language. Therefore, we used the original English version of the scales. This has now been clarified in the revised manuscript on page 7 lines 209 to 210;

“All data were collected in English language given that all the medical students in the present study were proficient in English because all their medical training is conducted in this language.”

---

## [Editor Report · Decision Letter 1]

16 May 2023

Problematic use of the internet, smartphones, and social media among medical students and relationship with depression: An exploratory study

PONE-D-23-01046R1

Dear Dr. Muwanguzi,

We’re pleased to inform you that your manuscript has been judged scientifically suitable for publication and will be formally accepted for publication once it meets all outstanding technical requirements.

Kind regards,

Giulia Ballarotto

Academic Editor

PLOS ONE
---

## [Editor Report · Acceptance letter]

18 May 2023

PONE-D-23-01046R1 

Problematic use of the internet, smartphones, and social media among medical students and relationship with depression: An exploratory study 

Dear Dr. Muwanguzi:

I'm pleased to inform you that your manuscript has been deemed suitable for publication in PLOS ONE. Congratulations! Your manuscript is now with our production department. 

Kind regards, 

on behalf of

Dr Giulia Ballarotto 

Academic Editor

PLOS ONE